# Evidence of Biparental Mitochondrial Inheritance from Self-Fertile Crosses between Closely Related Species of *Ceratocystis*

**DOI:** 10.3390/jof9060686

**Published:** 2023-06-19

**Authors:** Daniella van der Walt, Emma T. Steenkamp, Brenda D. Wingfield, P. Markus Wilken

**Affiliations:** Department of Biochemistry, Genetics and Microbiology, Forestry and Agricultural Biotechnology Institute (FABI), University of Pretoria, Pretoria 0028, South Africa; daniella.kramer@fabi.up.ac.za (D.v.d.W.); emma.steenkamp@fabi.up.ac.za (E.T.S.); markus.wilken@fabi.up.ac.za (P.M.W.)

**Keywords:** *Ceratocystis*, hybridization, PCR RFLP

## Abstract

Hybridization is recognized as a notable driver of evolution and adaptation, which closely related species may exploit in the form of incomplete reproductive barriers. Three closely related species of *Ceratocystis* (i.e., *C. fimbriata*, *C. manginecans* and *C. eucalypticola*) have previously been shown to hybridize. In such studies, naturally occurring self-sterile strains were mated with an unusual laboratory-generated sterile isolate type, which could have impacted conclusions regarding the prevalence of hybridization and inheritance of mitochondria. In the current study, we investigated whether interspecific crosses between fertile isolates of these three species are possible and, if so, how mitochondria are inherited by the progeny. For this purpose, a PCR-RFLP method and a mitochondrial DNA-specific PCR technique were custom-made. These were applied in a novel approach of typing complete ascospore drops collected from the fruiting bodies in each cross to distinguish between self-fertilizations and potential hybridization. These markers showed hybridization between *C. fimbriata* and *C. eucalypticola* and between *C. fimbriata* and *C. manginecans*, while no hybridization was detected in the crosses involving *C. manginecans* and *C. eucalypticola*. In both sets of hybrid progeny, we detected biparental inheritance of mitochondria. This study was the first to successfully produce hybrids from a cross involving self-fertile isolates of *Ceratocystis* and also provided the first direct evidence of biparental mitochondrial inheritance in the *Ceratocystidaceae*. This work lays the foundation for further research focused on investigating the role of hybridization in the speciation of *Ceratocystis* species and if mitochondrial conflict could have influenced the process.

## 1. Introduction

Hybridization is one of the various forces driving the evolution of new species [1]. It is defined as the crossing of genetically distinguishable groups that leads to the production of viable offspring [2]. This process can result in genetic admixture, which may promote adaptation if the hybrid progeny possesses combinations of beneficial alleles [3]. Through selection, individuals carrying such novel and beneficial allelic combinations may evolve into separate species, having greater virulence and adaptability than their parent species. This has been seen in the evolution of some plant pathogenic fungi, where hybridization led to the emergence of novel species or genotypes responsible for widespread devastation [1,3,4,5,6].

Fungal hybrids are often encountered in studies aiming to use the biological species concept for delineating species of *Ceratocystidaceae*, especially in *Endoconidiophora* and *Ceratocystis* [7,8]. In these fungi, intraspecific crosses and self-fertilizations (or selfings) cause the formation of ascomata (sexual fruiting bodies) with large, clear ascospore masses at the top of long ascomatal necks [7]. However, interspecific crosses between different *Endoconidiophora* species result in the formation of ascomata with distorted asci and ascospores and reduced ascospore viability [7]. Indeed, single-ascospore progeny of the few successful interspecific crosses of *Endoconidiophora* showed very poor growth, typical of interspecific hybrids [7].

In *Ceratocystis*, interspecific hybridization has been used to delineate species in the *C. fimbriata* complex [9]. A notable example is the study of three monophyletic lineages in the complex’s Latin American clade, which are, respectively, host-specialised to cacao (*Theobroma cacao*), sweet potato (*Ipomea batatas*) and sycamore (*Platanus* spp.) [9]. Sexual mating crosses were used to assess the existence of reproductive barriers between them. Here, crosses producing unusually small numbers of ascospores were taken as evidence of species divergence between the respective lineages, and host specialization likely mediated the process [9]. Accordingly, the cacao pathogen was described as *C. cacaofunesta,* while the sycamore pathogen was named *C. platani* [9].

Species in the genus *Ceratocystis* include many economically important plant pathogens that affect various notable tree and crop species [8,10,11]. The genus also includes the species *C. manginecans*, which represents an example of a plant pathogen whose evolution has been linked to hybridization [12]. This fungus causes disease on an unusually wide range of plant hosts, e.g., *Acacia mangium* [13,14,15], *Mangifera indica* [16], *Eucalyptus* species [17], *Punica granatum* [18], *Dalbergia sissoo* and *Prosopis cineraria* [19]. Most other *Ceratocystis* species affect single plant hosts. For example, the type species of the genus, *C. fimbriata* [20], causes black rot only on sweet potato [21,22,23,24,25], and the recently described *C. eucalypticola* appears to infect only *Eucalyptus* trees [26].

All *Ceratocystis* species undergo unidirectional mating-type switching during sexual reproduction [27,28,29,30]. A common trait of fungi utilizing this system is the natural presence of two sexual fertility types: self-fertile and self-sterile isolate types [27,29,31,32,33]. At the gene level, the mating-type (*MAT1*) locus of self-fertile isolates is characterised by the *MAT1-1* gene being split by the presence of the *MAT1-2* genes [29]. The region containing the *MAT1-2* genes is flanked by direct repeats [29], which support an as yet unknown recombination event that mediates permanent deletion of the *MAT1-2* genes from the locus [34,35,36]. Isolates in which this deletion has occurred are self-sterile and contain only the *MAT1-1* gene information [29]. However, the picture is somewhat more complex, as a second self-sterile type has also been described from several species in the *Ceratocystidaceae*, including *Ceratocystis* [9,37,38]. This second type of self-sterile isolate has only been generated under laboratory conditions, and the basis of its self-sterility remains unclear [9,29,37]. From what is known so far, they apparently encode an intact *MAT1* locus with all the requisite *MAT1-1* and *MAT1-2* gene information [29]. However, as with the *MAT1-2-*lacking self-sterile isolates, the *MAT1-2-*containing self-sterile isolates also remain mating competent and can act as a parent in sexual crosses [29].

Studies on interspecific hybridization in the *Ceratocystidaceae* have made use of compatible matings between *MAT1-2-*lacking self-sterile isolates and *MAT1-2-*containing self-sterile isolates [7,30,39]. This assured that the development of ascomata was due to true hybridization events and not the result of self-fertilization, as would have been the case when self-fertile isolates were used [29]. Such crosses between the two self-sterile isolate types of *C. fimbriata* and *C. eucalypticola* were investigated to determine whether reduced fertility of progeny could contribute to post-zygotic reproductive isolation [39]. Similarly, these isolate types were used in crosses of *C. fimbriata* and *C. manginecans* to study the phenotype of hybrid progeny [11,30]. In all these cases, the *MAT1-2-*lacking self-sterile isolates acted as the maternal parent based on the progeny’s inheritance of mitochondria [30,39]. Because none of these previous studies employed self-fertile isolates of *Ceratocystis* in their experiments, it is unclear how the various conclusions drawn relate to what might happen under natural conditions. Also, *MAT1-2-*containing self-sterile isolates are considered laboratory-based artifacts and may not be present in natural populations [7,9,30,39].

Therefore, the aim of this study was to determine whether self-fertile isolates, the most commonly encountered isolate type of most *Ceratocystis* species [27,29,36], are capable of hybridization. This was achieved by employing self-fertile isolates in crosses among the three closely related species *C. fimbriata*, *C. eucalypticola* and *C. manginecans* [8]. Based on the development of ascomata, the relative abundance of outcrossing and self-fertilization was estimated using PCR-based methods. We also used PCR to study mitochondrial inheritance in the progeny. The results of this study would contribute to the knowledge base on fungal hybrids and more specifically sets the scene for further studies into the role of hybridization in the speciation process of these *Ceratocystis* species.

## 2. Materials and Methods

### 2.1. Interspecific Hybridization of Ceratocystis Isolates

Self-fertile isolates of *Ceratocystis manginecans* (CMW46461), *C. eucalypticola* (CMW9998) and *C. fimbriata* (CMW14799) were obtained from the culture collection (CMW) of the Forestry and Agricultural Biotechnology Institute (FABI) of the University of Pretoria, South Africa. Isolates were maintained at 25 °C on MEA-TS medium, which contained 2% (*w*/*v*) Malt Extract Agar supplemented with Thiamine (100 mg/L) and Streptomycin (150 mg/L). Their self-fertile nature was evidenced by the presence of ascomata that bear ascospore drops (Figure 1).

To produce hybrid progeny, the three self-fertile isolates were paired in all possible combinations on petri dishes containing MEA-TS. Crosses were made by using a sterile needle for collecting a single ascospore drop from an ascoma of a particular parental isolate and placing the drop approximately 1 cm away from the drop obtained from another parent. Five plates for each of the three parental pairings were prepared and incubated in containers containing silica crystals at 25 °C for a period of two weeks. A zone of interaction between the isolates was defined as the area on the medium where the mycelium of the two *Ceratocystis* species came into contact, and it is in this region where hybridization was expected to occur. Within the zone of interaction, the ascospore drops from five randomly selected ascomata per plate were collected for each cross (i.e., twenty-five drops were collected per cross). Single spore drops were also collected from each side of every plate, away from the interaction zone, to serve as non-hybrid controls, as the ascomata produced in this region likely resulted from self-fertilization. All the collected spore drops were plated onto fresh MEA-TS medium and incubated for 21 days at 25 °C until the mycelia covered most of the plate. The entire mating experiment was conducted in duplicate.

### 2.2. DNA Extraction and PCR-Based Confirmation of Parental Self-Fertility

Genomic DNA was isolated from the original parental isolates and all cultures grown from the collected ascospore masses by using an extraction protocol based on hexadecyltrimethyl ammonium bromide (CTAB) [40]. Briefly, fungal tissue was scraped from the entire surface of each plate with a sterile scalpel blade and then vortexed in the presence of extraction buffer (0.2 M Tris, 1.4 M NaCl, 20 mM EDTA, 0.2 g/L CTAB) containing 2 or 3 glass beads. Mixtures containing the disrupted fungal tissue were incubated for 3 min at 100 °C and then 10 min on ice. Supernatant was collected via centrifugation for 5 min at 18,000 rcf, after which the nucleic acids contained within the aqueous phase were purified using chloroform:isoamylalcohol (24:1) extraction. Following overnight precipitation at −20 °C in the presence of 2.5 M ammonium acetate and 2 volumes of isopropanol, nucleic acids were recovered via centrifugation at 18,000 rcf for 10 min at 4 °C. The resulting pellets were washed with 70% ethanol, air-dried and suspended in sterile water.

To confirm self-fertility of the parental isolates, they were screened for the presence of the *MAT1-1-1* and *MAT1-2-1* genes. To achieve this, portions with particular sizes were amplified for the respective genes using primers (Table 1) previously designed for *C. fimbriata* [27] and also tested on related *Ceratocystis* species [39]. The target fragments were amplified using 25 µL PCR mixtures that contained 1 U of KAPA *Taq* DNA polymerase (Kapa Biosystems, Wilmington, MA, USA), 1 × KAPA *Taq* Buffer A, 0.4 mM of each primer, 0.25 mM of each dNTP and 90-150 ng of template DNA. A non-template control without any DNA template was also included in every reaction. An Eppendorf Thermocycler (Eppendorf AG, Mannheim, Germany) was used for amplification with the following parameters: 5 min at 95 °C followed by 35 cycles of 30 s at 95 °C, 30 s at 50 °C and 30 s 72 °C and a final elongation step of 72 °C for 7 min. Amplicon sizes were estimated using 1% agarose (SeaKem LE Agarose, Lonza, Rockland, ME, USA) through gel electrophoresis for 20 min at 150 V, after which the DNA stained with GelRed Nucleic Acid Gel stain (Biotium, Hayward, CA, USA) was visualised with an ultraviolet transilluminator.

### 2.3. Marker and Primer Design

Two sets of markers were developed in this study; one was designed using nuclear DNA regions to allow for differentiation of *C. fimbriata*, *C. eucalypticola* and *C. manginecans* from one another, while the second targeted their mitochondrial genomes. For this purpose, the genome sequences for the respective species were obtained from the database (https://www.ncbi.nlm.nih.gov/genome/ of the National Centre for Biotechnology Information (NCBI) using the accession numbers SGIO00000000, APWK03000000 and LJOA00000000 for, respectively, *C. manginecans* CMW46461, *C. fimbriata* CMW14799 and *C. eucalypticola* CMW9998.

The *C. fimbriata* genome had been previously annotated [41] and was used to design three sets of nuclear markers to use as a diagnostic for identifying the individual species. An in silico analysis of restriction enzyme (RE) digestion patterns was used to find intergenic regions with sequence variation that could be used in a PCR-based restriction fragment-length polymorphism (PCR-RFLP) analysis. This was conducted by randomly taking portions of the *C. fimbriata* genome and performing a BLAST comparison to the *C. manginecans* and *C. eucalypticola* genomes to find regions of sequence variation. Three variable regions were selected, aligned among the three species and then subjected to in silico digestion for identifying unique RE cut sites. Regions that were considered suitable as RFLP markers were then used as input into Primer3Plus [42,43] to design primers for PCR amplification of the respective regions (Table 1).

Mitochondrial genome sequences for the three species were compared as before to identify variable regions that could potentially be used to identify the parental origin of the mitochondria inherited by the progeny. The sequences of these variable regions together with their flanking sequences were then used as input into Primer3Plus to design primers that would allow for amplicons of different sizes for the three species (Table 1). In other words, this diagnostic method included three primer sets that could, when used in combination, identify the origin of the mitochondria.

### 2.4. Marker Analysis

The primers designed for the RFLP analysis were used to amplify the targeted marker regions using DNA extracted from the parental isolates, as well as from the progeny. Each 25 µL reaction mixture included 1 U of KAPA *Taq* DNA polymerase (Kapa Biosystems, Wilmington, MA, USA), 1 × KAPA *Taq* Buffer A, 0.4 mM of each primer, 0.25 mM of each of the dNTPs and 90-150 ng of template DNA. A non-template control without any DNA template was also included in every reaction. The PCR cycling conditions were as follows: an initial denaturation step of 3 min at 94 °C, followed by 25 cycles of 30 s at 94 °C, 30 s at 50 °C and 30 s at 72 °C and a final extension step of 7 min at 72 °C. The amplified PCR products were stained with GelRed and visualised on 1% agarose via gel electrophoresis after 20 min at 150 V.

RFLP analysis of the amplicons was performed using the appropriate high-fidelity enzymes (New England Biolabs, Ipswich, MA, USA) as per the manufacturer’s instructions. One unit of each RE and 1 × rCutSmart Buffer was added to the PCR reaction, which was then incubated for 1 h at 37 °C followed by an inactivation step at 80 °C for 20 min. GelRed was added to the inactivated digestion, and the full reaction was separated on a 1% agarose gel for 60 min at 90 V and visualised under UV light. The restriction profile of each spore drop culture was compared to the expected banding pattern for the individual parents to identify possible hybridizations. Progeny was scored as the result of hybridization if both banding patterns of the parents were present in a culture established from a single spore drop.

The primer sets that targeted mitochondrial genomes were amplified as described above, except that an annealing temperature of 52 °C was used. Following PCR, amplicons were visualised as described above, after which their sizes were used to identify the mitochondrial origin of potential hybrids from the *C. manginecans* × *C. eucalypticola* and *C. manginecans* × *C. fimbriata* crosses. However, primer set 1 produced almost the same size fragment in *C. fimbriata* and *C. eucalypticola*, which necessitated the use of sequence analysis for scoring mitochondrial origins of their progeny. To achieve this, the relevant amplicons were purified using Sephadex G50 columns (Sigma-Aldrich, St. Louis, MO, USA), and the amplicons were sequenced using feMitSeq_F primer, the BigDye Terminator Cycle Sequencing Kit v.3.1 (Life Technologies, Carlsbad, CA, USA) and an ABI3500xL Genetic analyser (Applied Biosystems, ThermoFisher Scientific, Waltham, MA, USA) at the DNA Sanger Sequencing facility, Faculty of Natural and Agricultural Science, at the University of Pretoria. The resulting electropherograms were visually inspected for diagnostic regions unique to the mitochondrial genome of *C. eucalypticola*.

## 3. Results

### 3.1. Interspecific Hybridization of Ceratocystis Isolates

On MEA-TS medium, the three parental isolates (i.e., *C. manginecans*, *C. eucalypticola* and *C. fimbriata*) all produced abundant ascomata, which served as an initial indication of their self-fertility. PCR amplification of the *MAT1-1-1* and *MAT1-2-1* genes from these cultures all produced amplicons of the expected sizes (i.e., 436 bp and 495 bp, respectively). Therefore, the presence of both the *MAT1-1-1* and *MAT1-2-1* fragments confirmed that the isolates were indeed self-fertile and had not shifted to self-sterility through unidirectional mating-type switching [27,34,35,44].

When mated with each other, all interspecific crosses produced some ascomata within the zone of interaction (Figure 2). However, the number of fruiting bodies present in this zone was consistently less than the number present in regions outside this region where selfing reactions likely occurred. Culturing of the spore drops collected in the zones of interaction from the various MEA-TS plates again resulted in cultures that abundantly produced ascomata.

### 3.2. Development of a PCR-RFLP Method Based on Nuclear Markers

Three sets of nuclear markers with suitable levels of polymorphism were used to design a PCR-RFLP method for differentiating *C. manginecans*, *C. eucalypticola* and *C. fimbriata* from one another. These were based on the sequence comparisons of three genes. In the case of nuclear marker 1, our primers targeted part of the noncoding region to the one side of the endoplasmic reticulum membrane protein 65. For marker 2, the primers targeted a gene encoding endothelin-converting enzyme-like 1. The region targeted by the primers for marker 3 corresponded to the beginning of a gene encoding bifunctional purine biosynthetic protein ADE1, as well as part of the non-coding region upstream from it. Laboratory testing of these markers yielded the predicted PCR-RFLP profiles for each of the parental species (Appendix A). Details regarding each of these markers are provided below.

The primers for nuclear marker 1 were designed to amplify a 692 bp fragment in all three *Ceratocystis* species. This fragment had a single *Hin*dIII cut site in all three species, as well as two *Pst*I cut sites in *C. eucalypticola* and *C. fimbriata* (Figure 3a). In silico digestions of *C. eucalypticola* and *C. fimbriata* with *Hin*dIII and *Pst*I, thus, produced four bands of sizes 35, 170, 190 and 290 bp. The lack of the second *Pst*I cut site in *C. manginecans* provided this fungus with a unique band pattern (35, 290 and 360 bp fragments; Figure 3d). Therefore, this marker could differentiate *C. manginecans* from *C. eucalypticola* and *C. fimbriata*.

Marker region 2 was designed around an 836 bp amplicon that would be amplified in all three *Ceratocystis* species (Figure 3b). In silico digestion with *Hin*dIII predicted the formation of two bands (100 bp and 740 bp) for *C. manginecans* and *C. fimbriata*, with neither containing *Pst*I cut sites (Figure 3d). In *C. eucalypticola*, this region had four *Hin*dIII and one *Pst*I cut sites (producing six fragments of sizes 55 bp, 85 bp, 90 bp, 100 bp, 110 bp and 400 bp, respectively). Therefore, this marker region could differentiate *C. eucalypticola* from *C. manginecans* and *C. fimbriata*.

The primers for nuclear marker 3 were designed to produce an amplicon of 908 bp. This marker would differentiate *C. fimbriata* from the other two species, as a unique *Pst*I cut site and common *Hin*dIII cut site produces three bands sized 560, 180 and 160 bp, respectively, when digested in silico with these restriction enzymes (Figure 3c,d). In *C. manginecans* and *C. eucalypticola*, digestion with *Hin*dIII and *Pst*I produces two bands of sizes 340 bp and 560 bp.

### 3.3. Identifying Hybrid Cultures Using PCR-RFLP Analysis of Nuclear Markers

PCR-RFLP analysis using the three polymorphic nuclear markers identified in this study allowed for differentiation of *C. manginecans*, *C. eucalypticola* and *C. fimbriata* isolates from one another and from cultures potentially representing hybrids of these species. If restriction profiles obtained from progeny cultures were identical to any of the three parental isolates, they were regarded as being the product of selfing (i.e., self-fertilization of the parent isolate). If the restriction profile represented a mixture of the profiles for both parents, the progeny culture was scored as being the product of a hybridization between the parents (example of one of the markers from a cross given in Figure 4).

#### 3.3.1. *C. fimbriata × C. eucalypticola*

For the cultures grown from 25 ascospore drops collected during the first round of interspecific crosses, marker 2 was successfully amplified in only 23 of the resulting cultures, despite multiple PCR attempts. Among these, 4 cultures showed the banding pattern of *C. fimbriata* only (indicative of *C. fimbriata* selfing), 3 showed the banding pattern of only *C. eucalypticola* (indicative of *C. eucalypticola* selfing) and 16 showed a banding pattern characteristic of both *C. fimbriata* and *C. eucalypticola*, indicating hybridization. Marker 3 was amplified from all 25 spore drop cultures, with the RFLP scoring 6 isolates as *C. fimbriata* selfings, 7 as *C. eucalypticola* selfings and 12 as potential hybrids (Appendix A). During the second round of crosses, marker 2 was successfully amplified from all 25 spore drop cultures, as was marker 3. However, neither marker region 2 nor 3 showed a selfing of *C. fimbriata*. Marker 2 showed 15 *C. eucalypticola* selfings and 10 hybridizations, while marker 3 showed 22 *C. eucalypticola* selfings and only 3 hybridizations (Appendix A). The spore drops collected as controls showed the expected banding pattern of the individual parent. Taken together (Table 2), these data showed that among the 50 mating events examined during the two rounds of crosses, 26 were scored involving hybridization and 24 as selfings (20 in *C. eucalypticola* and 4 in *C. fimbriata*).

#### 3.3.2. *C. fimbriata × C. manginecans*

For cultures grown from the 25 ascospore drops collected from the first round of crosses, both marker 1 and marker 3 were successfully amplified from 24 of the resulting cultures. Using nuclear marker 1, none of the cultures were scored as originating from hybridization, while 23 and 1 were scored as being the product selfing in, respectively, *C. fimbriata* and *C. manginecans*. With marker 3, we scored 3 of the cultures as having hybrid origins, while 1 and 20 cultures were scored as selfings of *C. manginecans* and *C. fimbriata*, respectively (Appendix A). For the second round of crosses, evidence of hybridization was detected only with marker region 1. Of the selfings scored with marker 1, the majority (20) originated from *C. fimbriata* with only a single selfing in *C. manginecans* detected. With marker region 3, we only detected selfings of *C. fimbriata*, except for one detected as a *C. manginecans* selfing event (Appendix A). Therefore, of the 48 mating events between *C. fimbriata* × *C. manginecans*, only 8 bore evidence of hybridization, while 40 were regarded as selfings (1 in *C. manginecans* and 39 in *C. fimbriata*) (Table 2).

#### 3.3.3. *C. manginecans × C. eucalypticola*

For the first round of *C. manginecans* × *C. eucalypticola* crosses, 25 spore drops were collected and grown on MEA-TS medium. Marker regions 1 and 2 were successfully amplified from 24 of the resulting cultures. Based on their PCR-RFLP profiles, 23 of these were scored as being the product of selfing of *C. eucalypticola* and 1 as a selfing of *C. manginecans* (Appendix A). In the second round of crosses, marker region 1 was successfully amplified from all 25 cultures, of which PCR-RFLP analysis suggested that 19 originated from *C. eucalypticola* selfings and 6 from *C. manginecans* selfings. With marker region 2, which was successfully amplified only for 22 of the spore drop cultures, 16 were scored as *C. eucalypticola* selfings and 6 as *C. manginecans* selfings (Appendix A). None of the ascospore drop cultures, thus, appeared to have originated from hybridization (Table 2).

### 3.4. Mitochondrial DNA Inheritance in Hybrid Cultures

Three primer sets were designed to amplify regions of the mitochondrial genome that are diagnostic for the three *Ceratocystis* species examined (Appendix A). The mitochondrial genome was not annotated in the genome assembly, and, therefore, it was not known if these regions were within a gene. However, these markers were used to study the inheritance of mitochondria in the cultures identified as having originated from hybridization events between the crosses examined here (Table 2). All eight of the hybrid cultures obtained from the crosses between *C. fimbriata* and *C. manginecans* yielded two distinct amplicons using primer set 1. Of these, one corresponded to the 560 bp fragment diagnostic for *C. manginecans* and the other to the 1890 bp fragment of *C. fimbriata*. Similarly, for primer set 2, an amplicon was produced from all hybrid cultures, although the differences in size were too small to assign parental origin. This result was unexpected, as the in silico analysis indicated that no amplicon should be possible in the *C. manginecans* parent, and the marker was not used to assign origin to the hybrids. With primer set 3, all of these hybrid cultures contained the 560 bp diagnostic for *C. fimbriata*. In other words, the mitochondria of both these parental strains were inherited by their hybrid progeny (Figure 5).

For all 26 of the ascospore drop cultures collected from the crosses between *C. fimbriata* and *C. eucalypticola*, PCR with primer set 3 yielded the 560 bp fragment expected in *C. fimbriata*. PCR with primer set 1 yielded an 1890 bp fragment expected for both parents. Therefore, to determine their origins, the amplicons were sequenced, which in most cultures showed the presence of *C. fimbriata* mitochondria (Appendix A). However, the sequence data also suggested the presence of *C. eucalypticola* mitochondrial DNA in at least 6 of the 26 hybrid cultures. This was apparent from the presence of two sequencing profiles in the region containing the 9 bp deletion characteristic of *C. eucalypticola* (Appendix A). Therefore, PCR combined with sequence information provided evidence of both parents’ mitochondrial DNA being present in some of the ascospore drops originally collected from the ascomata produced during the *C. fimbriata* × *C. eucalypticola* cross.

## 4. Discussion

Here, we used a novel approach to demonstrate that self-fertile isolates of different *Ceratocystis* species can hybridize and that their progeny often bears the mitochondria of both parental species. Although the importance of hybridization during adaptation and evolution is widely recognized [45,46], fungal hybridization is not completely understood, and few studies have explored the level to which sexual reproductive barriers and species boundaries are permeable [47,48]. By employing cultures grown from single ascospore drops obtained from crosses between *C. fimbriata* and *C. manginecans* and between *C. fimbriata* and *C. eucalypticola,* we showed, for the first time, that self-fertile isolates of closely related *Ceratocystis* species can sexually reproduce, forming viable progeny with mitochondrial inheritance from both parents. All previous attempts at this “forced” outcrossing by mating self-sterile isolates lacking *MAT1-2* with self-sterile isolates containing *MAT1-2* [30,39]. In contrast to *MAT1-2*-lacking self-sterile isolates that are present among the progeny of self-fertilizations [7,29,36,39], *MAT1-2*-containing self-sterile strains are not encountered in nature [7,9,30,39]. The use of these two types of self-sterile isolates might, thus, skew the data obtained from sexual mating experiments. Therefore, our experiments with self-fertile isolates provided more conclusive support to the leaky reproductive barriers previously reported between some of these species [39,49,50]. Additionally, these findings provide a strong basis from which to explore the roles of decreased fertility and fitness as post-zygotic reproductive barriers in these fungi [39].

The fact that crosses between *C. manginecans* and *C. eucalypticola* did not produce hybrid progeny was unexpected. Both of these species were interfertile with *C. fimbriata* and, based on rigorous phylogenetic analyses, the three species are very closely related [8]. However, neither *C. manginecans* nor *C. eucalypticola* have been widely studied, and we likely have an incomplete picture of their ecology and distribution. *C. eucalypticola* has only been reported from *Eucalyptus* species in South Africa [26,51], while *C. manginecans* has been reported on various hosts and from various localities in the Middle East, Malaysia, India, Brazil, and South China [13,14,17,19,20,28]. Therefore, meaningful explorations on the basis of the apparent reproductive barrier between *C. manginecans* and *C. eucalypticola* require analyses of broad collections of isolates that are representative of both their geographic distribution and host range [52,53].

The presence of PCR markers from both parents in the mitochondria of the examined hybrid progeny is interesting, as these organelles typically show a uniparental inheritance pattern in fungi [54,55,56] that is distinct to that of nuclear genes [57,58]. In plants and animals, uniparental inheritance of mitochondria is mostly facilitated by pre-zygotic processes related to gamete anisogamy, where the maternal gamete is larger and contains many more mitochondria compared to the paternal gamete [54,55,56,59]. Not much is known about the regulation of mitochondrial inheritance in fungi, but various possible mechanisms might be involved, ranging from the physical position of mitochondria within dividing cells through to enzymatic degradation of additional types of mitochondria or mitochondrial DNA [60]. Nevertheless, the paradigm of uniparental mitochondrial inheritance was previously reported for interspecific [30,39] and intraspecific [30,39] crosses of *Ceratocystis*. For interactions between conspecifics, one *Ceratocystis* parent’s mitochondria or mitochondrial DNAs are purged from the progeny [60,61,62]. In the case of hybrid crosses, however, the results of these initial studies were likely impacted by the use of *MAT1-2*-containing self-sterile strains, especially when evaluating reproductive barriers. In other words, the reduced fitness associated with these types of sterile isolates likely extends to their mitochondria, as our data clearly showed that mitochondria of both parental species can end up in the hybrid progeny.

Biparental inheritance of mitochondria is often associated with hybridization, where the condition of harbouring more than one type of mitochondrion is referred to as heteroplasmy [58,63]. This condition is due to a breakdown in the coordinated set of mechanisms governing the typical uniparental inheritance of these organelles [58]. Biparental inheritance has been reported before in crosses between the plant pathogens *Verticillium dahlia* and *V. nonalfalfae* [52,64], as well as among various species of yeast and human pathogens [65,66]. In the current study, the crosses between *C. fimbriata* and *C. manginecans* and the crosses between *C. fimbriata* and *C. eucalypticola* all produced heteroplasmic progeny, and their analysis using genetics and comparative genomics methodologies would be invaluable, not only for understanding species boundaries but also for developing a holistic knowledge framework for why and how these fungi emerged as pathogens.

Another intriguing explanation for the presence of mitochondria from both parents in the hybrid progeny could stem from the sexual strategy employed by *Ceratocystis*. During unidirectional mating-type switching, apparently self-fertile isolates produce ascospores that are either self-fertile or self-sterile [27,34,35,44]. In *Chromocrea spinulosa*, another species capable of unidirectional mating-type switching, there is evidence that nuclei of opposite mating-type co-exist within a single cytoplasm, and karyogamy followed by meiosis would produce a mixture of self-fertile and self-sterile ascospores [34]. Whether this mechanism is at play in *Ceratocystis* is currently not known. However, it could be speculated that in the *Ceratocystis* hybrid crosses presented here, each parent contributed two nuclei to the fertilization event that produced an ascoma—one self-fertile nucleus and one self-sterile nucleus in a single cell with its own mitochondria. The parent that acted as the maternal parent (i.e., the cell that donated the self-sterile nucleus [30,39]) would then dictate which cell donates cytoplasm and mitochondria during karyogamy. This would result in the mitochondria of a single ascospore originating from one parent only. However, among the many ascospores in a spore drop, there could be variations in which cell acted as the maternal parent and, therefore, acted as the mitochondrial donor. Genetic typing of complete spore drops (as conducted here) would then produce a profile associated with biparental mitochondrial inheritance. Our future work will seek to explore this hypothesis further by examining the mitochondria present in single isolates derived from these self-fertile *Ceratocystis* crosses.

Our novel use of cultures established from single ascospore drops obtained from distinct ascomata for detecting hybridization and to track mitochondrial inheritance is reminiscent of a technique previously used to score self-fertile and *MAT1-2*-lacking self-sterile isolates in the progeny of *Ceratocystis* drops [67]. Because PCR may be associated with amplification bias [68], it is conceivable that an ascospore drop produced from a hybrid interaction is characterized by only one parent profile. Similarly, any segregation bias in the marker regions or differences in the viability of ascospores contained within a spore drop would present as an unequal profile in the PCR-RFLP analyses or would produce a varied signal in the electropherogram profiles of the Sanger sequencing results. Although it would be tempting to use these to make conclusions on the contribution of each parent to the offspring (e.g., imbalanced genome contributions or the ratio of mitochondria per parent), the use of complete spore drops in the assays would make these data unreliable. The nature of these assays likely resulted in an underestimation of the possible number of hybridization events examined, which was further compounded by the use of only two marker regions to characterise hybrids. Additionally, each mating used only one individual from each of the three species, although fertility within populations of *Ceratocystis* has been shown to vary widely [30]. Despite this, neither PCR bias nor the choice of isolates would have impacted our conclusions regarding hybridization and mitochondrial inheritance in the three fungi examined. Our methodology, therefore, introduces a feasible strategy with which to identify hybrids and trace mitochondrial inheritance across *Ceratocystis* and the broader *Ceratocystidaceae*.

## Figures and Tables

**Figure 1 jof-09-00686-f001:**
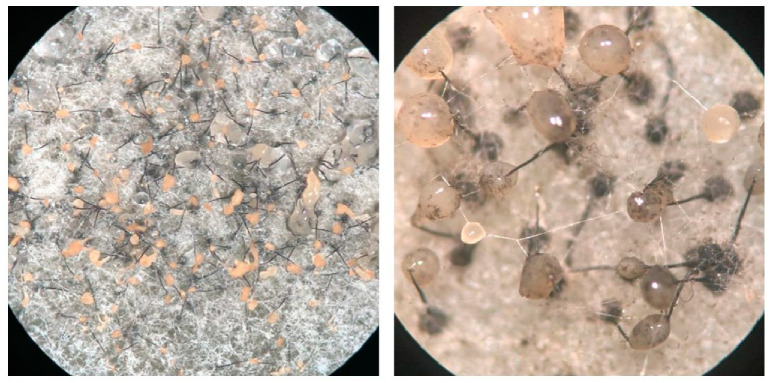
*Ceratocystis* growing on MEA media, showing the sexual fruiting bodies (ascomata). Ascomata have round, dark bases with long necks, at the top out of which slimy masses of ascospores exude.

**Figure 2 jof-09-00686-f002:**
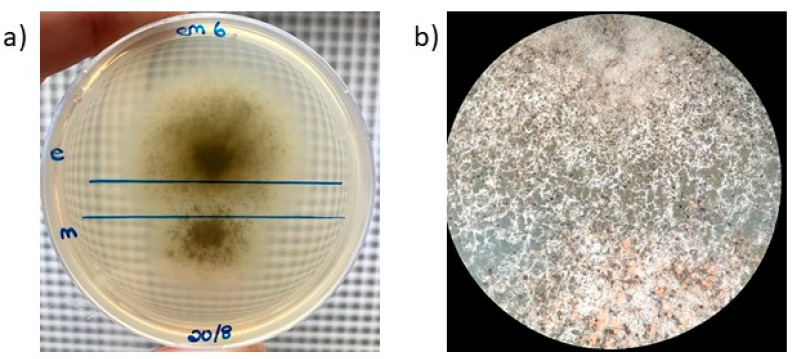
A cross of *C. fimbriata* and *C. eucalypticola* following incubation on MEA-TS medium. (**a**) Parallel blue lines show the zone of interaction. (**b**) Close-up of the interaction zone.

**Figure 3 jof-09-00686-f003:**
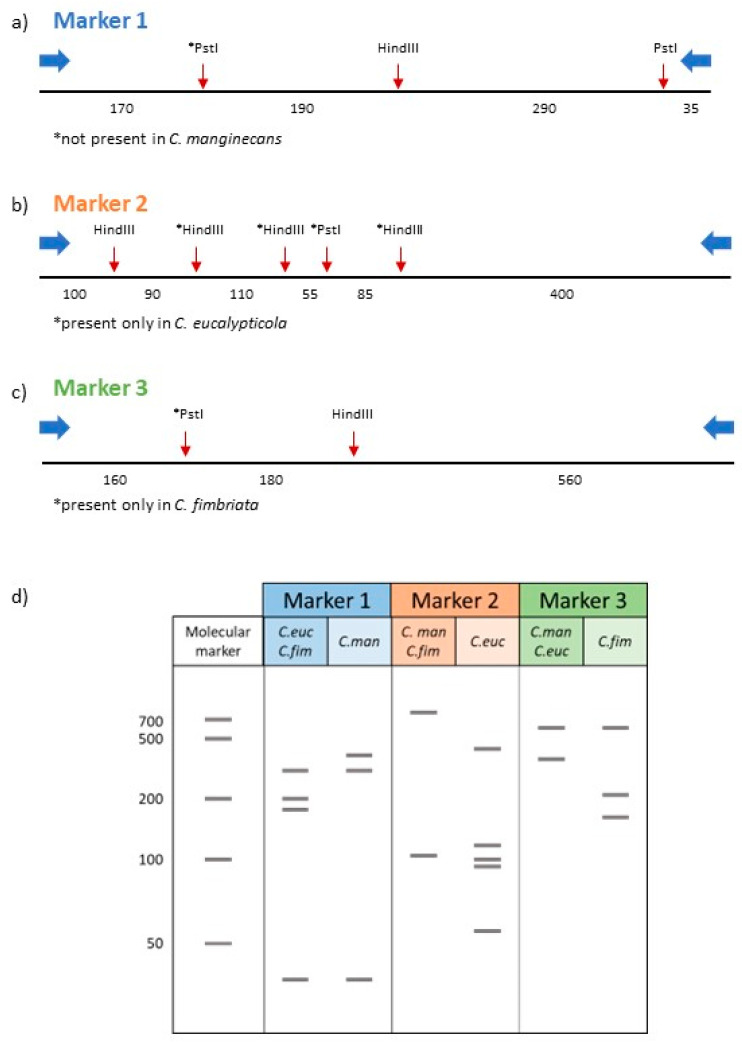
In silico PCR-RFLP analysis of the three nuclear genomic regions containing *Hin*dIII and *Pst*I restriction enzyme cut site variation in the examined *Ceratocystis* species. (**a**) Marker region 1 contains a *Pst*I cut site that is absent only in *C. manginecans*. (**b**) Marker region 2 contains a *Pst*I cut site present only in *C. eucalypticola*. (**c**) Marker region 3 has *Pst*I cut site unique to *C. fimbriata*. (**d**) Predicted profiles following digestion of the three marker regions with *Hin*dIII and *Pst*I (molecular marker in base pairs).

**Figure 4 jof-09-00686-f004:**
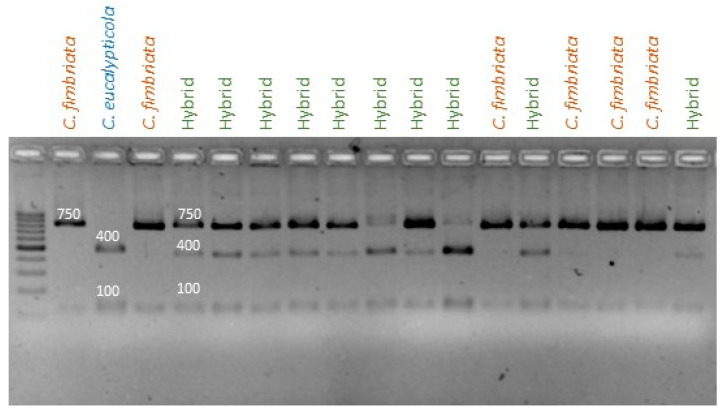
An example the PCR-RFLP profiles produced for nuclear marker 2 applied to a *C. fimbriata* × *C. eucalypticola* mating. The amplicon of 836 bp of marker 2 (primers marker2_R and marker2_F) is double digested using restriction enzymes *Hin*dIII and *Pst*I. In both species, the presence of a band at approximately 100 bp indicates successful digestion. A 750 bp band is unique to *C. fimbriata*, while in *C. eucalypticola* the presence of a unique cut produces a band at 400 bp. Therefore, only lanes that clearly contain all three bands (100 bp, 400 bp and 750 bp) following digestion are considered representative of a hybrid.

**Figure 5 jof-09-00686-f005:**
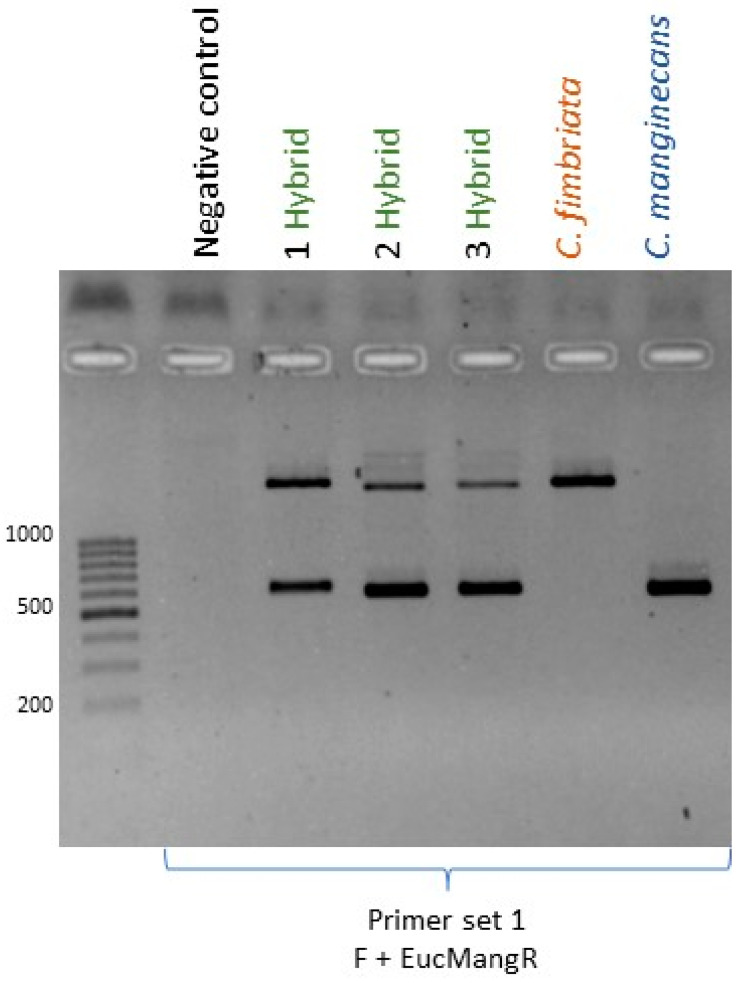
An agarose gel showing biparental mitochondrial inheritance in an interspecific cross of *C. fimbriata* × *C. manginecans*. The amplicons shown were produced using mitochondrial primer set 1 (Primer F and EucMangR), which can effectively distinguish *C. manginecans* from the other two species. Amplification in *C. fimbriata* yields a large band of around 1800 bp, while in *C. manginecans* a smaller band of around 560 bp is produced. In a spore drop from a hybrid cross between these two species, two distinct bands of approximately 1800 bp and 560 bp are seen, confirming the presence of mitochondria from both parents.

**Table 1 jof-09-00686-t001:** List of primers used in this study and notes on their specific usage in the various experiments.

Primer Name *	Sequence (5′ → 3′)	Size (bp)	Notes
marker1_R	TTTCTGCTGTCCACACCTTG	692	Nuclear primer set 1. Region amplified from all three species but restriction enzyme (RE) digestion produces unique banding pattern for *C. manginecans*.
marker1_F	TTTCTGCTGTCCACACCTTG		
marker2_R	GACCGCATGGTTGAGGTTAC	836	Nuclear primer set 2. Region amplified from all three species but RE digestion produces unique banding pattern for *C. eucalypticola*.
marker2_F	GTTCATGATGCCATCGACAC		
marker3_R	GTGTAGCCGTCGGAAAATGT	908	Nuclear primer set 3. Region amplified from all three species but RE digestion produces unique banding pattern for *C. fimbriata*.
marker3_F	TGTTGGATGGGCTGTATTGA		
Cf_Mt_F1	GAAGTGCCTTCGCTTTAT	436	Targets the *MAT1-1-1* gene in all three species.
Cf_Mt_R1	GACCGCGATTCTAACCAAAA		
MAT1-2-1F	AAGATGCTCTTTAATACCCACCA	495	Targets the *MAT1-2-1* gene in all three species.
MAT1-2-1R	TGCCGCTAATAAGCTAGGAA		
Primer F	AATTGGATCTTCAACGACTAAAC	562 1886 1895	Mitochondrial primer set 1. Amplicon of 562 in *C. manginecans*, and amplicons of 1886 and 1895 bp in *C. eucalypticola* and *C. fimbriata*.
EucMangR	GGGCTCTGTTAGTCTCTGCA		
Primer F	AATTGGATCTTCAACGACTAAAC	387 387	Mitochondrial primer set 2. Amplicons of 387 and 387 bp in *C. eucalypticola* and *C. fimbriata*.
EucR	TCCGCGATCATCCATTTCTC		
FimMito2F	CCTGCATCTCGTCCTATCGT	559	Mitochondrial primer set 3 amplifies only in *C. fimbriata*.
FimMito2R	TCGCCTTTCAAGTTCCATGC		
feMitSeq_F	CATTTGGGGGCTTTTTGTAA	n/a	Primer used to sequence amplicons produced with mitochondrial primer set 1 in order to score the region of variability between *C. eucalypticola* and *C. fimbriata*.

* Primer F was used in both mitochondrial primer sets 1 and 2.

**Table 2 jof-09-00686-t002:** Summary of the results for the PCR-RFLP and mitochondrial PCR analyses for the two rounds of crossing experiments conducted in this study (see Appendix A for detail).

Interspecific Cross ^1^	Nuclear Markers ^2^		Mitochondrial Marker
*C. fimbriata* × *C. eucalypticola* (50 spore drops)	Both markers	15	Biparental mitochondria confirmed in 6 hybrid spore drops ^3^
	Single marker	11	
	Total number of hybrids	26	
*C. fimbriata* × *C. manginecans* (50 spore drops)	Both markers	0	Biparental mitochondria confirmed in 8 hybrid spore drops
	Single marker	8	
	Total number of hybrids	8	
*C. manginecans* × *C. eucalypticola* (50 spore drops)	Both markers	0	n/a ^4^
	Single marker	0	
	Total number of hybrids	0	

^1^ A total of 50 spore drops were collected from each of the crosses, 25 per round over 2 rounds. The complete spore drop was grown, and DNA was extracted for RFLP analysis. ^2^ Two markers were evaluated per cross, with number of spore drops showing a hybrid pattern at both makers (Both markers) or only one marker (Single marker) indicated. ^3^ The six positive results reported here is based on the Sanger sequencing results. See text for details. ^4^ Could not be tested as no hybrids were detected.

## Data Availability

Publicly available datasets were analysed in this study. These data can be found at https://www.ncbi.nlm.nih.gov/, using the information provided in the manuscript.

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
