# Peer review of "Evidence of Biparental Mitochondrial Inheritance from Self-Fertile Crosses between Closely Related Species of Ceratocystis"

_jof, 2023, doi:10.3390/jof9060686_

Round 1
Reviewer 1 Report
This work begins to build the plant-associated fungi, Ceratocystis, as a model for hybridization studies. The authors design molecular markers (nuclear and mitochondrial) to distinguish between 3 Ceratosystsis species and perform mating experiments to determine the ability for these species to hybridize. The authors find evidence for hybridization between Cf x Ce and Cf x Cm but not Ce x Cm. Interestingly, the authors found evidence for biparental inheritance of mtDNAs in some of the hybrids, expanding our understanding of organelle inheritance in fungi.
While the molecular reporters aren’t robust (often a lack of a band is taken as proof for a species) and some controls are missing (see below), this work contributes to our tool set for these fungi that will be useful for future work, and contributes to our understanding of hybridization and organelle inheritance.
1. A major contribution of this work is presenting molecular markers that can be used for species determination. The predicted in silico amplification and digestion patterns are presented but only a sentence (line 248-249) that states that the predicted nuclear PCR-RFLP profiles were obtained for each of the parent species. I recognize that not all agarose gels will be publication quality, but it would be nice to include an actual gel of these controls as a proof of principal, including digested and undigested amplifications. This would make it easier to interpret the gels that are presented, and would make it easier for others to potentially use these markers.
2. Fig 4:
a. Categorization of hybrid status is based on the presence/absence of the 400 bp fragment. How confident can you be that the lack of the 400bp fragment is Cfim and not a partially digested Ceuc amplification? (the first point could maybe address this?)
b. In a hybrid with balanced genome contributions from both parents, I would expect that the numbers of the 700 bp and 400 bp fragments to be equivalent (though the 700 bp band should provide almost 2x the amount of signal as the 400 bp band). The ratio of band intensities varies in the putative hybrids, where often the 700 bp fragment seems to be in higher abundance than the 400 bp band (examples: hybrids 2-6) though sometimes this is reversed (eg. Hybrid 7). Do the authors have any insight here? Could this be imbalanced genome contributions, or do you think this is technical artifact? Could imbalanced genome contributions fit with the proposed mating strategies described in the discussion (lines 433-452)?
c. Sometimes a spore is marked as a parental (eg. Sample 9 Ceuc) when there is a faint band corresponding to the alternate parent. At what point are you confident in calling something a hybrid vs. a selfed offspring?
d. Fig 4 shows a sample gel that leads to the data in Fig S1. This figure should be better described. Presumably this is marker 2, digested with Pst1? How do the lanes in the gels in Fig 4 match with the data shown in Table S1? Do fe 1A-B from Table S1 match samples 1-5 in Fig 4?
3. Summarizing the hybridization data into Table 2 is an excellent idea, though the current version of this table is difficult to interpret. How can 24 “selfing” crosses show both only one and both nuclear markers? The authors should consider an updated version to this table that indicates the numbers of each possible outcome (for example, the numbers of each selfed parent, the numbers of times an offspring showed both species for both markers vs 1 and 1. The mitochondrial amplifications could be better summarized: 6 of 26 hybridization events led to amplification of the mtDNA marker from both parents (in Cf x Ce). How many times were only the Cf or Ce markers amplified? This could provide some insight into preferred mtDNA transmission.
4. The electropherogram in Fig S2 suggest that there are roughly equal amounts of Cf and Ce sequences in the hybrid. Was this seen in all the hybrids? Could this suggest that each species mtDNA is maintained as separate mtDNAs (no recombination) (and in roughly equal proportions)? This would be in contrast to saccharomyces yeasts, where biparental mtDNA inheritance and recombination and fixation for a single mitotype is the norm (yeast don’t maintain mtDNA heteroplasmy), and suggests that numerous mechansims for mtDNA maintenance exist in Ascomycetes.
Author Response
Reviewer 1 comments:
- The reviewer suggested that we include an agarose gel of the RFLP profiles for each parent species as proof of principle, which includes digested and undigested amplifications.
We agree that agarose gels showing these results would provide more clarity on the RFLP markers designed in this study. Therefore, two agarose gels were added as supplementary Figures S1 a) and S1 b) to complement the extensive analysis of the restriction profiles provided (section 3.3). Figure S1 a) shows the PCR amplicons and restriction profiles for each nuclear marker set across all three species, while Figure S1 b) compares the restriction profiles for all three nuclear markers across all three species.
- The reviewer had several helpful suggestions relating to Figure 4.
- Firstly, the reviewer queries our confidence in categorising hybrid status, and whether the absence of a 400 bp fragment could be considered evidence of fimbriata selfing or perhaps only partially digested C. eucalypticola amplification.
Based on this and other comments from the reviewer, it became apparent that both the Figure and the caption could be improved. Therefore, as a first step the Figure in question was trimmed to show less samples but be clearer. In addition, the Figure legend was also extensively edited. From the updated Figure legend, it should now be clear how the assignment of hybrids was done, and therefore explain how the absence of a certain fragment impacts the results.
- Based on the varied intensities of the bands shown in Figure 4, the reviewer suggested that imbalanced genome contributions could be involved. However, the reviewer also noted that this might be a technical artifact.
Although we agree that there is a realistic chance that imbalanced genome contributions might be at play in this study, it was not the intention of this work to evaluate this. The use of spore drops for the RFLP analyses creates background noise in the PCR amplifications that may be related to PCR bias or the viability of individual ascospores, making it impossible for us to confidently draw conclusions from the intensities of the amplicons alone. A short discussion on this was added to the last paragraph of the discussion (lines 485 to 491).
- The reviewer’s third point also related to the confidence we had in assigning parentage to any spore drop.
This comment very closely matched that provided in point a above, and we have improved the legend of the Figure to address this. However, the parentage was assigned incorrectly for the specific sample referred to, and this oversight on our side has been corrected.
- The reviewer had several queries based on Figure 4, and how it relates to Figure S1.
Although these two Figures are unrelated, we could see how, after reading the comments from the reviewer, there was some confusion. The main concern is that Figure 4 does not relate to Figure S1 (now Figure 5). Therefore, to address this concern both Figure legends have revised to add more clarity. Based on the comments of the reviewer, we moved Figure S1 to the main text to replace Figure 5, and the original Figure 5 now forms part of the supplementary material (Figure S2).
- The reviewer suggested that Table 2 should be updated to make it easier to interpret.
We agree with the reviewer that Table 2 was difficult to interpret. To address this, the Table was redesigned to make it easier to understand the data. The Table now only shows information related to the hybrid spore drops, clearly sets out the results per cross over both repeats and provides information for both the nuclear markers and mitochondrial markers clearly and distinctly. Several explanatory footnotes were added to aid in understanding the information. We are sure that this will make the Table easier to interpret.
- The reviewer noted that, in Figure S2 (now Figure S4) there appears to be roughly equal amounts of fimbriata and C. eucalypticola sequences in the presented electropherogram. They then asked if this was consistent across all the hybrids.
We thank the reviewer for raising this point and note that it is in principle similar to the comment raised in comment 2b (above). Although only a selection of hybrid spore drops from the C. fimbriata and C. eucalypticola cross was subjected to Sanger sequencing, we did not evaluate the electropherograms in that amount of detail. We believe that the use of spore drops makes it difficult to reliably interpret the electropherograms as they relate to parental contribution. We have added text to the discussion (lines 485 to 491) to address this.
Reviewer 2 Report
The manuscript "Evidence of biparental mitochondrial inheritance from self-fer- tile crosses between closely related species of Ceratocystis" reports the interspecific crosses between fertile isolates of these three species of Ceratocystis (i.e., C. fimbriata, C. manginecans and C. eucalypticola). This work lays the foundation for further research about the role of hybridization in the speciation of Ceratocystis species. I think this mannuscript is worth publishing.
After reading the manuscript again, I have several questions about the reliability of the results. The manuscript showed surprisingly high proportion of hybridization (26 of 50) of C. fimbriata × C. eucalypticola, and the hybrids inherited mitochondrial of both parents which is conflicting with our existing cognition. Therefore, the authors should be caution that the so-called hybrids were rather contamination. The authors are encouraged to supply more analysis to exclude the contamination and add more experiment to prove their conclusion, before consideration to publish.
1 The authors applied only 1 PCR-RFLP marker to analyze each of the hybridization, it is a very weak evidence. More markers or other methods such as chromosome painting, genome sequencing, were needed to exclude false positives
2 Did the hybrids generated morphological traits inherited from both parent?
3 The F2 separation analysis were needed to prove the hybrids.
Minor comments:
1 line 211-212,the “C. manginecans X C. eucalypticola and C. 211 manginecans X C. fimbriata crosses” should be “C. manginecans × C. eucalypticola and C. 211 manginecans × C. fimbriata crosses”
2 line 293, C. fimbriata X C. eucalypticola should be C. fimbriata × C. eucalypticola; the X should be replaced with × in line 310, 324 and 372 et al.
English is understandable, however, there are small syntax error. Some sentences are too lengthy, it is recommended to be concise and clear.
Author Response
Reviewer 2 comments:
- The reviewer noted a high proportion of hybrids in one of the crosses (26 out of 50 for the fimbriata × C. eucalypticola cross) and raised the possibility that these could be the result of contamination. It was also mentioned that the finding of biparental mitochondrial inheritance is contrary to the existing cognition.
Although we appreciate the comment, we do not agree that this is a high proportion of hybrids. As explained in lines 118-121 and Figure 2, we specifically targeted a region of the cross that would be more likely to contain hybridization events, and it is also known that hybrids can form between C. fimbriata and both C. manginecans and C. eucalypticola (as stated in lines 81-89). As such, we were surprised that more hybrids were not detected.
Similarly, biparental inheritance in fungi has been reported before as discussed in line 452-456. In Ceratocystis species this was never studied as the crosses used restricted the possibility of biparental mitochondrial inheritance (discussed in lines 89-94, as well as lines 410-419). Therefore, we believe that the evidence presented here clearly illustrates the potential of biparental mitochondrial inheritance.
With regards to the comment on the contamination, we are unsure as to the exact source of contamination that the reviewer is referring to. One potential source of contamination could be from external fungi. This would not be an issue in this study as the primers were highly specific and were custom designed using the genome sequences (section 2.3). Additionally, no-template PCR controls were included in all reactions (this has now been explicitly stated in lines 151-152 and line 192-193), and all amplicons were of the expected size. Additionally, the contamination referred to could stem from conidia or mycelia of the parents that may be collected together with the ascospore drops. We do not believe this to be the case. This technique has been used before in a related Ceratocystis species (references in lines 483-486) with success. The ascospore drops of these species are large and suspended on a long neck (as seen in Figure 1), making collection easy. Additionally, if contamination was truly a concern one would expect a 100% hybrid rate across all the crosses. This was not the case, with hybrid rates varying between 0% and 50% per cross (Table 2) despite designing the study to specifically find hybridization.
- The reviewer questioned the use of a single PCR-RFLP marker per cross to show that there was hybridization.
We are not sure where the confusion arose, but in all cases two markers were used per cross to identify hybridization (discussed in sections 3.3.1, 3.3.2, and 3.3.3). We do concede that in some cases (11 of the 26 C. fimbriata × C. eucalypticola hybrids, and all 8 of the C. fimbriata × C. manginecans hybrids) hybrids were identified by only a single marker. However, apart from hybridization there is no way to explain the presence of two distinct profiles for a single marker in a spore drop, and Figure S1 clearly shows the expected profiles for hybrids. In addition, it is distinctly possible that the methods used in this study underestimates the amount of hybridization as set out in the discussion (lines 484-495).
- The reviewer requested that the morphological traits of the hybrids should be discussed, and that an F2 separation analysis be included.
In this study, single spore drops were used for all analyses. Therefore, no single individuals were ever created in the F1 generation. Therefore, no information on the morphology of the offspring could be presented, and no F2 separation analysis is possible. However, we take these suggestions under advisement for future projects.
- The reviewer recommends that the “X” between manginecans X C. eucalypticola, for example, be changed to a “×”.
We agree with this suggestion and the changes were made throughout the paper.
- The reviewer noted minor syntax errors in the paper as well as suggested more concise and clear sentences.
Several minor errors were identified and corrected, and where possible lengthy sentences were shortened for clarity.
Round 2
Reviewer 1 Report
The revised manuscript has been greatly improved such that it is easier to follow the data and figures. I am now confident that the conclusions are appropriate and that this makes a nice contribution to the Ceratocystis model system and our understanding of fungal hybridization.